# Artemisinin Attenuates Amyloid-Induced Brain Inflammation and Memory Impairments by Modulating TLR4/NF-κB Signaling

**DOI:** 10.3390/ijms23116354

**Published:** 2022-06-06

**Authors:** Xia Zhao, Xiaosu Huang, Chao Yang, Yizhou Jiang, Wenshu Zhou, Wenhua Zheng

**Affiliations:** 1Center of Reproduction, Development & Aging and Department of Pharmacology, Faculty of Health Sciences, University of Macau, Macau SAR 999078, China; yb77625@um.edu.mo (X.Z.); yb97634@um.edu.mo (C.Y.); yb77642@um.edu.mo (Y.J.); yb87618@um.edu.mo (W.Z.); 2Institute of Translation Medicine, Faculty of Health Sciences, University of Macau, Macau SAR 999078, China; 3Hangzhou Medical College, Hangzhou 310000, China; 4School of Nursing, Guangdong Pharmaceutical University, Guangzhou 510006, China; 15915736152@163.com

**Keywords:** Alzheimer’s disease, Artemisinin, neuroinflammation, microglia, cognitive disorder

## Abstract

The abnormal immune response is an early change in the pathogenesis of Alzheimer’s disease (AD). Microglial activation is a crucial regulator of the immune response, which contributes to progressive neuronal injury by releasing neurotoxic products. Therefore, finding effective drugs to regulate microglial homeostasis and neuroinflammation has become a new AD treatment strategy. Artemisinin has potent anti-inflammatory and immune activities. However, it is unclear whether Artemisinin contributes to the regulation of microglial activation, thereby improving AD pathology. This study found that Artemisinin significantly reduced amyloid beta-peptide 1–42 (Aβ_1–42_)-induced increases in nitric oxide and reactive oxygen species and inflammatory factors in BV2 cells. In addition, Artemisinin inhibited the migration of microglia and prevented the expansion of the inflammatory cascade. The mechanical studies showed Artemisinin inhibited neuroinflammation and exerted neuroprotective effects by regulating the Toll-like receptor 4 (TLR4)/Nuclear factor-kappa B (NF-κB) signaling pathway. Similar results were obtained in AD model mice, in which Artemisinin administration attenuated Aβ_1–42_-induced neuroinflammation and neuronal injury, reversing spatial learning and memory deficits. The anti-inflammatory effect of Artemisinin is also accompanied by the activation of the TLR4/NF-κB signaling pathway in the animal model. Our results indicate that Artemisinin attenuated Aβ_1–42_-induced neuroinflammation and neuronal injury by stimulating the TLR4/NF-κB signaling pathway. These findings suggest that Artemisinin is a potential therapeutic agent for AD.

## 1. Introduction

Alzheimer’s disease (AD) is the most common form of dementia in the elderly [1]. According to statistics, there were nearly 50 million patients with AD in 2018, and the number is expected to exceed 152 million by 2050, of which 13.8 million will be patients over the age of 65, which will cause a huge burden for families and societies [2]. So, it is urgent to develop new effective drugs for AD treatment.

Neuroinflammation is an important pathophysiological feature of AD [3]. Microglia are the primary immune cells of the central nervous system (CNS), playing a pivotal role in neuroinflammation. It accounts for 5–20% of all glial cells in the mammalian adult brain [4]. Microglia activation is a significant feature of the CNS inflammatory response [5]. In the brains of AD patients, ~80% of the “amyloid plaque” structure is covered by activated microglia [6]. Activated glial cells can cause the release of proinflammatory cytokines such as tumor necrosis factor (TNF)-α, interleukin (IL)-1β, and IL-6 and induce the production of neurotoxic such as nitric oxide and reactive oxygen species [7]. The expression of these proinflammatory cytokines causes neuroinflammation and further neuronal damage [7,8]. Besides, the inflammatory environment may induce tau hyperphosphorylation and promote the formation of neurofibrillary tangles [9]. Studies have shown that Aβ can induce microglial activation, significantly impacting the central nervous system [10]. Therefore, finding effective drugs to regulate microglial homeostasis and neuroinflammation may be a new strategy for AD treatment.

Artemisinin is extracted from the plant Artemisia annua and was discovered by Chinese pharmacist Tu Youyou in 1971. Tau won the Nobel Prize in physiology or medicine in 2015 for the discovery of this drug [11]. Artemisinin is the first-class antimalarial drug used for decades [12]. Artemisinin and its derivatives (Arts) have anti-inflammatory effects [13,14]. We also showed that they have a neuronal protective effect [15]. Artemisinin can easily cross the blood–brain barrier with low toxicity [16]. Therefore, Artemisinin has many properties of an ideal CNS drug. However, few studies on its regulation effect on microglial homeostasis and its effect on microglia-mediated inflammation on neuron injury in CNS diseases. Our preliminary study showed that pretreatment with Artemisinin could inhibit the release of Aβ-induced inflammatory cytokines (such as TNF-α, IL-6, and IL-1β) in BV2 cells. These inflammatory cytokines can induce apoptotic neuronal death or amplify the local inflammatory response, resulting in synaptic dysfunction or neuronal loss [17]. The innate immune receptor Toll-like-receptor 4 (TLR4), localized on the surface of microglia, is an important defense receptor against invading microorganisms. Mutation of TLR4 strongly inhibits microglial and monocytic activation by aggregated Alzheimer amyloid peptides resulting in a significantly lower release of the inflammatory products IL-6, TNFα, and nitric oxide [18]. Therefore, TLR4 is a critical innate immune receptor involved in neuroinflammatory and neurodegenerative diseases [19]. The activation of TLR4 results in the activation of downstream mediators, including the transcription factor nuclear factor (NF)-κB, which increases the production of proinflammatory molecules [20]. Moreover, the inhibition of the TLR4/NF-κB signaling pathway suppressed the expression of IL-6 and TNF-α [21]. This study aimed to establish a microglial inflammatory cell model and animal disease model induced by Aβ to observe the effect of Artemisinin on the microglial inflammatory response and to determine its mechanism. This knowledge could be helpful for the prevention and treatment of Alzheimer’s disease.

Aβ_1–42_ has substantial neurotoxicity, and it is one of the leading causes of neuronal degeneration and death in AD patients. The hippocampus is critical for learning and memory [22], and injecting Aβ_1–42_ into the hippocampus induces pathological changes related to AD, such as Aβ_1–42_ deposition, neuronal damage, and cognitive dysfunction [23]. In this study, we injected Aβ_1–42_ into the mice hippocampus to establish an AD model and studied the effect of Artemisinin in AD model mice and microglia cells. Our study provides a new idea for the treatment of AD, and also opens up new directions for the clinical application of Artemisinin. 

## 2. Results

### 2.1. Establishment of Inflammation Model in BV2 Cells 

Amyloid plaque is an important pathological feature of AD, and it can activate microglia and trigger inflammation response [24]. Therefore, we chose to use Aβ_1–42_ to establish a microglial inflammation model. We treated BV2 cells with different concentrations of Aβ_1–42_ for 24 h and then detected the expression of the significant inflammatory factors TNF-α and IL-1β. Our results showed that the expression of inflammatory factors was significantly increased after Aβ_1–42_ treatment starting from 4 μM, and to a maximum of 8 μM (Figure 1A–C). Next, we checked the effect of Aβ_1–42_ on BV2 cell membrane integrity. We tested the release of LDH in the supernatant after treating BV2 cells with different concentrations of Aβ_1–42_ for 24 h. Our results showed that Aβ_1–42_ significantly increased LDH release starting from 1 μM, and increased with the increase in Aβ_1–42_ treatment concentration (Figure 1D). Based on the above results, we choose 4 μM Aβ_1–42_ to establish the inflammation model for follow-up experiments.

### 2.2. Artemisinin Decreased the Production of ROS and iNOS in BV2 Cells

Under healthy circumstances, microglial cells constantly extend and retract to inspect the environment. When the central nervous system is infected or injured, microglia are activated, somatic cells are swollen, and synapses disappear and undergo a series of changes in gene expression and function [25]. Overactivated microglia release cytotoxic substances, such as ROS and iNOS. To test the anti-inflammatory effects of Artemisinin, we tested the production of ROS and iNOS in microglia and observed microglia morphological changes. Our results showed that Artemisinin treatment significantly reduced Aβ_1–42_-induced ROS and iNOS production (Figure 2A–C). ROS is an important mediator of COX-2 expression [26]. Therefore, we also used Western blotting to detect the expression of COX2 and iNOS. Our results showed that Aβ_1–42_ treatment increased the expression of COX2 and iNOS, while Artemisinin treatment decreased the expression of COX2 and iNOS (Figure 2D,E). These data demonstrate that Artemisinin treatment significantly decreased oxidative stress in BV2 cells.

### 2.3. Artemisinin Reduced the Release of Inflammatory Factors and Inhibited the Migratory Ability of BV2 Cells 

Activated microglia release various biologically active substances, cytokines, neurotransmitters, and cytotoxic substances. They participate in the initiation of inflammation, secretion of proinflammatory cytokines, and exhibit a series of changes in gene expression and function [27]. To further verify the anti-inflammatory effect of Artemisinin, we tested the expression of inflammatory factors. We first detected the release of the inflammatory factors TNF-α and IL-1β with the ELISA assay (Figure 3A,B). Obtained results showed that Aβ_1–42_ treatment significantly caused the release of TNF-α and IL-1β, while Artemisinin treatment significantly reduced the release of TNF-α and IL-1β. We then further verified this result by using Western blotting (Figure 3C,D). We found that Artemisinin significantly reduced the increased expression of inflammatory factors caused by Aβ_1–42_. These results indicate that Artemisinin can reduce the neuroinflammation caused by Aβ_1–42_ in BV2 cells.

The transcription factor NF-κB regulates multiple aspects of immune functions and serves as a key mediator of inflammatory responses. NF-κB induces the expression of various proinflammatory genes, including those encoding cytokines and chemokines, and also participates in inflammasome regulation [28]. Western blotting was used to evaluate the expression of NF-κB p65 signaling pathway markers in BV2 cells. Compared with the control group, TLR4/p-NF-κB/p-IκBα was significantly increased in the Aβ_1–42_ group. Conversely, Artemisinin decreased the expression level of TLR4/p-NF-κB/p-IκBα compared with that in the Aβ_1–42_ group (Figure 3E,F), suggesting that the effect of Artemisinin may be through TLR4/NF-κB signaling pathway.

Microglia migrated and were recruited around Aβ plaques to exert immune function [29]. Activated glial cells release inflammatory cytokines, which amplify the inflammatory response by activating and migrating to other cells [30]. We explored the effect of Artemisinin on the migration of microglia. We found that the microglia from the Aβ_1–42_ treatment attracted more migrating microglia compared to the control. However, after Artemisinin treatment, the number of migrating microglia caused by Aβ_1–42_ was significantly reduced (Figure 3G,H). Together, these data demonstrate that Artemisinin treatment significantly decreases neuroinflammation and inhibit the expansion of the inflammatory response cascade.

### 2.4. Microglia Conditioned Medium Regulates PC12 Cell Viability and Apoptosis

BV2 cells pretreated with Artemisinin for 2 h were stimulated with/without 4 μM Aβ for 24 h, and the conditioned culture supernatant was collected; PC12 cells were plated into three groups: CTL group, Aβ_1–42_ group, and Aβ_1–42_ + ART group, after adding BV2 serum to stimulate PC12 cells for 24 h, the viability and apoptosis of cells were detected. We found that Artemisinin-pretreated microglia medium reduced PC12 cell apoptosis (Figure 4A,B) and increased PC12 cell viability (Figure 4C). These data demonstrate that Artemisinin protects neuronal cell apoptosis by improving the inflammatory environment.

### 2.5. Artemisinin Reduced the Excessive Activation of Astrocytes in Primary Cultured Neurons

Astrocytes are primarily activated secondary to microglia [31], which is also related to the maintenance and persistence of the inflammatory response. GFAP is the main intermediate filament protein exclusive to astrocytes. We further checked the number of neuronal cells in primary cultured neurons. We found that Aβ_1–42_ treatment significantly reduced the number of neuronal cells and increased the number of astrocytes, while Artemisinin treatment increased the number of neuronal cells and decreased the number of astrocytes (Figure 5A–C). It showed that Artemisinin protected neuronal cells from apoptosis while reducing the excessive activation of astrocytes.

### 2.6. Artemisinin Improved the Cognitive Impairment of AD Model Mice

To further investigate the biological effects of Artemisinin, 30 male AD model mice were used. We injected AD model mice with 5 mg/kg Artemisinin daily for one month. Then, we analyzed the effect of Artemisinin on the cognitive function of C57 model mice by using the Morris water maze (MWM) test. The experimental setup and representative curves are shown in Figure 6A,B. We found that the average escape latency of the model group was significantly higher than that of the WT group, proving that our model was successfully established. The average escape latency of the Artemisinin-treated group was significantly lower than that of the model group (Figure 6C,D). These results show that Artemisinin improves the learning ability of the model group. On the next fifth day, we removed the platform. We evaluated the percentages of time each group of mice crossed the platform position and spent searching for the target quadrant (Figure 5E,F). The data showed that compared with the model group, the Artemisinin-treated group had more platform position crossing time, and the percentage of search time in the target quadrant increased, indicating that Artemisinin can improve the memory function of model mice. The above results indicate that Artemisinin can improve the cognitive impairment of AD model mice.

### 2.7. Artemisinin Reduced Neuronal Cell Damage in the AD Model Mice 

We used HE staining, Nissl staining, and TUNEL staining to detect the damage of brain neuronal cells in each group of mice. The results of HE staining showed that the neuronal cells of WT mice showed typical tissues with round and clear nuclei, while the cell morphology of model mice showed the contraction of nuclei. Artemisinin treatment improved the pathological characteristics (Figure 7A). Consistent with the observations in HE staining, a significant decrease in the relative number of Nissl-positive neurons was observed in the cerebral cortex of model mice, and Artemisinin treatment increased the relative number of Nissl-positive neurons (Figure 6B and Figure 7A). TUNEL staining results showed that the number of damaged neurons in the model group was significantly higher than that in the WT group, and Artemisinin treatment reduced the number of apoptotic neurons (Figure 7A,C). The above results indicate that Artemisinin can reduce neuronal cell damage in C57 model mice.

### 2.8. Artemisinin Reduced Neuroinflammation and Inhibited NF-κB Signaling in the AD Model Mice 

Neuroinflammation accompanied by abnormal activation of astrocytes and microglia has usually been observed in patients with AD and AD mouse models [32,33]. To further explore the effect of Artemisinin on inflammation in C57 model mice, we tested the expression of GFAP (astrocyte marker) and Iba1 (microglia marker). As expected, the number of GFAP-positive cells in the model mice was significantly higher than that in the WT group. Artemisinin treatment significantly reduced the number of GFAP-positive cells and Iba1-positive cells (Figure 8A–C), suggesting that Artemisinin has anti-inflammatory effects. We further used Western blotting to detect the protein expression of GFAP and Iba1. Our results showed that, compared to the WT group, GFAP and Iba1 in the model group were significantly increased, while the Artemisinin treatment group had significantly reduced expression of GFAP and Iba1 compared to the model group (Figure 8D,E). To further investigate the anti-inflammatory effect of Artemisinin, we verified the release of inflammatory factors. We found that treatment with Artemisinin reduced the release of inflammatory factors (Figure 8F,G). Consistent with the results obtained in the cells, Artemisinin inhibited the TLR4/NF-κB65 signaling pathway in C57 model mice (Figure 9A–C). The above results indicate that Artemisinin can improve the inflammatory response in the brains of C57 model mice. The anti-inflammatory effect of Artemisinin may be achieved by regulating the TLR4/NF-κB65 signaling pathway.

## 3. Discussion

In the past decades, significant efforts have been made to investigate the pathogenesis of AD and many hypotheses have been suggested. However, drugs based on these hypotheses have not shown significant clinical benefit so far. Microglia are the resident immune cells of the brain and the first responders to various abnormal environments [34]. Previous studies showed a correlation between AD and microglia activation [35]. Excessive activation of microglia has been proven to be one of the main mechanisms of AD pathogenesis [36]. Activated microglia release a variety of inflammatory factors, which leads to damage to hippocampal neurons and ultimately to cognitive dysfunction [37]. Drugs that improve microglial homeostasis and neuroinflammation are being considered as potential therapies for AD. 

In the present study, we explored the possibility that Artemisinin stabilizes microglia homeostasis and improves neuroinflammation, ameliorating AD pathology. We found that 4 μM Aβ_1–42_ induces excessive activation of BV2 cells and the release of inflammatory factors. Treatment with Artemisinin can inhibit the transfer of activated glial cells. ELISA was used to detect the expression level of cytokines in the supernatant of BV2 cells. We found that Aβ_1–42_ stimulated the production of TNF-α and IL-1β, while Artemisinin treatment significantly reduced their production. Further studies showed that Artemisinin can reduce the levels of ROS and iNOS in BV2 cells, alleviate oxidative stress in glial cells, and thereby further aggravate the inflammatory response of BV2 cells. 

Currently, the mechanism by which Artemisinin suppresses microglia neuroinflammation remains unknown. TLR4 is the most studied member of the TLR family, which responds to inflammation and mediates inflammatory signal transduction, such as nuclear factor kappa B (NF-κB) [38,39]. In a recent study, high expression of TLR4 was found in microglia [40]. Inhibition of TLR4 reduces inflammation and apoptosis [41]. In addition, several studies have also found that the expression of TLR4 and proinflammatory cytokines are increased in AD mice and AD patients [42,43]. These observations indicate that TLR4 plays a key role in regulating inflammation, and TLR4 inhibition can prevent inflammatory damage in the pathogenesis of AD [44]. NF-κB is a dimeric signaling complex that can stably exist in the cytoplasm after binding to IκBα [45]. In response to external stimuli, IκBα can be phosphorylated, causing IκBα to separate from NF-κB and causing nuclear translocation of NF-κB to initiate the transcription of certain inflammatory genes, leading to the production of inflammatory factors [46]. We found that the protective effects of Artemisinin were achieved by inhibiting the NF-κB signaling pathway in microglia. A previous study based on a transgenic mouse model of amyloid precursor protein has shown that Aβ can continuously activate the inflammatory response, including excessive activation of glial cells and activation of cytokines [47]. In our study, we found that TLR4 and NF-κB were highly expressed in BV2 cells after Aβ_1–42_ treatment, and these effects could be reversed by pretreatment with Artemisinin. The inflammatory cascade triggered by Aβ activation of astrocytes also plays an important role in the pathogenesis of AD [48]. Aβ deposition in the brain activates astrocytes and causes the release of cytokines. These cytokines accelerate the activation of astrocytes, Aβ deposition and the formation of fibrillary tangles through intracellular effects, self-effects, and proximity effects [49]. In addition, Aβ can up-regulate the expression of inflammatory factors and the production of Aβ after activating astrocytes [50], and in vivo and in vitro experiments have shown that Aβ can activate astrocytes and stimulate astrocytes to secrete proinflammatory factors [51]. GFAP is a primary intermediate filament protein that is specific to astrocytes [52]. Increased GFAP is usually considered to be an indicator of gliosis or a comparatively slow-developing index of neural damage linked with old age and the onset of AD pathology [53]. 

Then, we injected Aβ_1–42_ into the hippocampus of C57 mice to establish a model to study the potential role of Artemisinin in the C57 mouse model. The Morris water maze test showed that treatment of mice injected with Aβ_1–42_ with Artemisinin can significantly improve learning and memory, and reduce neuronal degeneration caused by the injection of Aβ_1–42_. The TUNEL staining results showed that Artemisinin reduced the number of apoptotic cells. In addition, after Artemisinin treatment, the markers of astrocytes (GFAP) and microglia (Iba-1) and inflammatory factors such as IL-1β, IL-6 and TNF-α were inhibited. The analysis of related pathways shows that the effect of Artemisinin may be achieved through the inhibition of the NF-κB signaling pathway.

To sum up, we demonstrated that the BV2 cells showed reduced levels of interleukin (IL)-6, tumor necrosis factor-α and IL-1β after Aβ_1–42_ treatment. Further studies suggested that in the Aβ_1–42_-induced inflammatory model, the protein levels of nuclear factor κ-light chain enhancer of activated B (NF-κB) p65 and Toll-like receptor 4 (TLR4) were increased, while Artemisinin pretreatment suppressed the activation of the TLR4/IκBα/NF-κB pathway. The data obtained show that Artemisinin reduces the release of inflammatory factors by inhibiting the TLR4/IκBα/NF-κB pathway (Figure 10).

## 4. Materials and Methods

### 4.1. Reagents and Chemicals

Analytical grade Artemisinin was purchased from Meilunbio (Dalian, China). Dimethyl sulfoxide (DMSO), Dulbecco’s modified Eagle’s medium (DMEM) and BSA were procured from Sigma (St. Louis, MO, USA). MTT, JC-1 and Hoechst 33258 were purchased from Molecular Probes (Eugene, OR, USA). Pierce BCA Protein Assay Kit were ordered from Thermo Fisher Scientific (Rockford, IL, USA). Annexin V-FITC/PI Apoptosis Detection Kit was obtained from BD Biosciences (San Diego, CA, USA). Fetal bovine serum (FBS) and 0.25% Trypsin were purchased from Life Technologies (Grand Island, NY, USA). DCFH-DA reagent and RIPA lysis buffer were ordered from the Beyotime Institute of Biotechnology (Shanghai, China). Penicillin/Streptomycin was purchased from Gibco. The detailed information of the antibodies is shown in Table 1. The Aβ_1–42_ sequence information is shown in Table 2. 

### 4.2. Animal and Treatment 

C57BL/6J (C57) mice were obtained from the Animal Research Core of the University of Macau. All animal experiments were performed under guidelines accepted by the University of Macau Animal Ethics Committee. The animal housing conditions were regulated, and animals were housed under controlled temperature (24–26 °C), humidity, and lighting (12-h light/dark cycle). Water and food were readily accessible.

C57 stereotactic injection of Aβ_1–42_ into the hippocampus model: A total of 30 mice were used in this experiment. C57 mice (six weeks old, male, *n* = 30, weight 22–25 g) were randomly segregated into three groups (10 mice/group): control, model, model + 5 mg/kg ART. ART was dissolved in 2% DMSO in PBS. One month before modeling, the ART + model group was intraperitoneally injected with 5 mg/kg ART, and the control and model groups were injected with the same volume of PBS. Then, the model group and the ART + model group were injected with Aβ_1–42_ (10 μg) into the hippocampus by a stereotaxic instrument, and the control group was injected with the same volume of PBS into the hippocampus. Experiments were performed seven days after modeling. 

### 4.3. Stereotaxic Injection 

After four weeks of Artemisinin treatment, the mice were weighed and anesthetized by intraperitoneal injection of 6% chloral hydrate at a dose of 0.6 mL/100 g. The anesthetized mice were fixed in a stereotaxic device, and the skin on the top of the head was prepared and sterilized with medical alcohol. A 1.5 cm long opening was made in the midsagittal of the head, and the soft tissue was wiped off with cotton moistened with 1×PBS to expose the bregma. The mouse was positioned according to the stereotactic map of the mouse to ensure that the mouse was placed on the same horizontal line before and after. The injection point was determined according to the corresponding coordinates and marked with a marker. We injected 2μL of the aggregated form of Aβ_1–42_ (5 μg/μL) into the hippocampus using a microsyringe within five minutes.The needle was kept in place for approximately five minutes after the injection, and the needle was slowly withdrawn within two minutes. The skin was sutured, and the wound was disinfected with alcohol to prevent infection. The coordinates of the hippocampus were as follows: the previous halo is zero, the rear opening is 1.9 mm, the side opening is 1.2 mm, and the needle insertion depth is 2.0 mm. As previously mentioned, Aβ_1–42_ was allowed to aggregate by incubation at 37 °C for seven days before use [54]. No mice died during this experiment.

### 4.4. Morris Water Maze (MWM)

The Morris water maze (MWM) test is a classic behavioral experiment that tests learning and memory ability [55]. The MWM was procured from ZS Dichuang (Beijing, China) having a xeye Aba video tracking system. The MWM is a circular pool (120 cm in diameter and 60 cm in height with white bottom and wall) with a white circular platform (diameter 8 cm; height 30 cm) submerged 1 cm under the water surface. Food-grade titanium dioxide (100 g) was added into the tank to whiten the water so that the mouse cannot visually recognize the platform. And the temperature was maintained at 25 °C. On each side of the wall of the quadrants, distinct colored papers were placed as visual positional hints. The video tracking camera was staged onto the ceiling directly above the center of the pool to monitor the subject’s swimming parameters.

For every training session, the mice were released to the maze back-to-back from four random points of the tank and were permitted to look for the platform for 60 s. In case the mouse did not find the platform within 60 s, they were securely directed to it. The time and movement route of the mouse to find the platform were recorded. Mice were kept in a drying cage with a heating system during the test interval. The above operation was repeated for the second, third, and four days. The average latency in the four-day navigational experiments of each group of mice was measured as an indicator for judging the learning ability of each mouse. On the fifth day, the platform was removed from the water maze, and the mice were allowed to swim freely for 60 s for space exploration experiments. The number of crossings of the platform and the residence duration of each mouse in each quadrant of the platform were recorded. All data acquisition and processing were carried out by the Morris water maze image automatic monitoring and processing system (ZS Dichuang, Beijing, China).

### 4.5. Tissue Preparation

After the MWM test, the mice were anesthetized with pentobarbital sodium (50 mg/kg) and the blood was sampled from the eyeballs for detection of serum TNF-α and IL-1β. Then, 1×PBS (pH7.4) was transcranial flushed in the mouse. Brains were quickly collected and dissected on ice, and the left hemisphere brain tissues were immersion fixed with 4% paraformaldehyde overnight and embedded in paraffin following the standard procedures, which can be stored for a long time [56]. For Immunohistochemical (IHC) and Immunofluorescence (IF) analyses, 5 µm thick sections were prepared. The cortexes of right hemisection were cut open on ice and kept at −80 °C for Western blot assay.

### 4.6. Immunohistochemistry and Immunofluorescence

Immunohistochemistry (IHC) is used to localize/visualize the expression of proteins in a mounted tissue section using protein-specific antibodies (H. Zhang et al., 2016). The brain tissue was sliced into 5 μm slices using Manual Rotary Microtome Basic Instrument (Leica RM2235, Nussloch, Germany). After routine dewaxing hydration, antigen retrieval was carried out by immersing in 0.01 M citrate buffer solution. This was followed by incubation with 3% H_2_O_2_ for 15 min to remove endogenous peroxidase activity. After blocking with 10% bovine serum albumin (BSA) for 1 h, the primary antibody was added dropwise to slices, which were stored overnight at 4 °C. The next day, the brain tissue sections were then incubated with the second antibody for 60 min and followed by DAB color development.

IImmunofluorescence (IF) is a method that uses a known fluorescein-labeled antibody as a probe to detect the target antigen in the tissue or cell to be tested, and the formed antigen-antibody complex binds to fluorescein, which can be observed under a fluorescence microscope. The brain tissue was embedded in OCT (optimal cutting temperature) compound. The brain was sliced into 20 μm slices using a low-temperature thermostat (Leica CM3050, Nussloch, Germany). Each section was washed with 1xPBS for three times and then blocked with 10%BSA for 1 h at room temperature. After that, tissue sections were incubated with the primary antibody in PBS containing 1% BSA at 4 °C overnight. On the following day, sections were incubated with the appropriate secondary antibody for 1 h at room temperature in the dark. Nuclei were counterstained with DAPI, and the images were acquired with a Nikon A1 confocal microscope.

### 4.7. Nissl Staining

Nissl staining was used to detect the surviving neurons. Nissl is large and large in number, indicating that nerve cells synthesize proteins with strong functions; on the contrary, when nerve cells are damaged, the number of Nissl bodies will decrease or even disappear. The embedded brain tissue was cut into 5 μm thin slices and followed by dewaxing in xylene and dehydrating by alcohol gradients. For each group, sections were stained with Nissl staining solution (C0117, Beyotime, Shanghai, China) for 10 min at 37–50 °C and then washed, color-separated, dehydrated, hyalinized. After staining, slices were dehydrated by fractional alcohols, trans parented with xylene and mounted with Neutral gum [57].

The cells which contained a clear body and nucleus were counted as surviving neurons in the sections of the hippocampus. The survival index was measured following this formula: survival index (%)  =  (number of surviving neurons/total number of neurons) ×  100%. 

### 4.8. BV2 Cell Line Culture

BV-2 is a type of microglial cell derived from C57/BL6 mice. The BV2 cells are immortalized by v-raf/v-myc carrying J2 retrovirus and are used to examine brain inflammation [30,58]. Hence, these cell lines were chosen for our study. 

BV2 cells were cultured in 75-cm^2^ flasks in DMEM supplemented with 10% heat-inactivated FBS and 0.1% penicillin/streptomycin at 37 °C with a 5% CO_2_ humidified atmosphere. The medium was replaced every 2–3 days, and cells were subcultured once 80–90% confluency was reached. After digestion with 0.25% trypsin, cells were collected by centrifugation at 1000 rpm for 5 min and resuspended in fresh medium. Cells were seeded into 96-well, 12-well or six-well plates and grown overnight. Adherent cells were used for further experiments.

### 4.9. Primary Neuronal Cells

Newborn C57BL/6 mice (within 24 h) were procured from the animal facility of the University of Macau. The whole body was disinfected with 75% alcohol and the brain was surgically removed and stored into cold HBSS (calcium- and magnesium-free) balance solution. The whole hippocampal region was dissected using a glass rod that was bent on both sides. The hippocampus was cleared of the blood and the mixed blood vessels by washing three times with HBSS. Then, the hippocampus was chopped into 1 mm^3^ pieces using scissors, and after washing three times with HBSS the tissue was digested with 0.125% trypsin at 37 °C for 15 min. The enzymatic digestion was stopped with 10% FBS and 5 mL of neurobasal A (Gibco, Carlsbad, CA, USA) was added to the digested hippocampal tissue in a 15 mL centrifuge tube. The turbid tissue supernatant was collected in another 15 mL centrifuge tube and centrifuged at 1000 rpm for 10 min. The resulting cell pellet was resuspended in Neurobasal A/B27 (Gibco, Carlsbad, CA, USA), seeded in poly-D-lysine -treated plates at a density of approximately 1 × 10^5^ cells/mL and incubated for growth at 37 °C in a 5% CO_2_ humidified atmosphere.

### 4.10. TUNEL Assay

Cell apoptosis was determined by the TUNEL assay performed using a TUNEL kit (C1098, Beyotime, Shanghai, China) following manufacturer’s instructions. Processed samples were incubated in a TUNEL reaction mixture (50 μL) for 60 min at 37 °C under dark conditions. The color was developed in the DAB coloring solution. Brown is apoptotic cells positive for TUNEL staining. Apoptosis index in the cortex area was measured using this formula: apoptosis index (%)  =  (apoptotic neurons/total neurons) × 100%.

### 4.11. Measurement of Intracellular ROS Levels

Intracellular reactive oxygen species (ROS) production was assessed by Cell ROXs Deep Red Reagent or DCFH-DA reagent. Briefly, cells grown in 96-well plates were incubated with 1 μM Aβ_1–42_ with or without pretreatment of the artemether. Cells were then incubated with Cell ROXs Deep Red Reagent or DCFH-DA reagent (5 μM) in fresh DMEM for 1 h in the dark. The cells were then washed with 1xPBS. Fluorescence was measured using a high content screening system (ArrayScan VTI, Thermo Fisher Scientific, Rockford, IL, USA) at 640 nm excitation and 665 nm wavelength for Cell ROXs Deep Red Reagent and 488 nm excitation and 525 nm wavelength for DCFH-DA reagent. Semi-quantitative ROS levels were normalized as a percentage of the control group [59].

### 4.12. ELISA 

After drug treatment, the expression levels of TNF-α and Il-1β in the cell supernatants or hippocampal homogenates were measured using ELISA kits according to the manufacturer’s instructions. Briefly, the samples and the testing solution were added to the wells, mixed gently, and incubated for 60 min at 37 °C. Then, the liquid was discarded, and washing buffer was added to each well, and the samples were washed three times. HRP conjugate was added, the samples were incubated for 20 min at 37 °C, the liquid was discarded, and the samples were washed five times. The substrate solution was added, and the samples were incubated in the dark for 20 min at 37 °C. Finally, stop solution was added to each well to stop the reaction. The optical density values were measured at 450 nm by a microplate reader within five minutes, and standard curves were plotted. Both standards and samples were measured in triplicate.

### 4.13. Transwell Assay

For microglial migration evaluation, Transwell assays were performed. Briefly, BV2 cells (1 × 10^5^) were seeded in the blank (FBS-free DMEM with Artemisinin) in the upper chamber of a Transwell. The lower chamber was filled with DMEM supplemented with 10% FBS. After 24 h of incubation, the cells in the upper chamber were removed, and the cells that had invaded the membrane were fixed with 100% methanol for 20 min and stained with crystal violet for 20 min. Images were acquired through crossed polarizers under a microscope.

### 4.14. Western Blot 

The shredded mouse brain tissue was blended in RIPA buffer containing protease inhibitors. The tissue sample was disrupted by sonication, and then centrifuged at 12,000× *g* for 20 min at 4 °C, and the supernatants were taken out in separate tubes. The protein concentration of each sample was determined with a BCA Protein Assay kit. The same amount of proteins were electrophoretically separated using polyacrylamide gel electrophoresis (PAGE) and transferred onto polyvinylidene fluoride (PVDF) membranes at 200 mA for 2 h. The PVDF membranes containing protein bands were blocked with 5% skimmed milk at room temperature for 1 h, and incubated with primary antibodies overnight at 4 °C. The following day, the membranes were incubated with horseradish peroxidase (HRP)-conjugated secondary antibodies (1:2000; CST) for 1 h at room temperature. The specific protein bands were seen using the Bio-Rad Gel Doc XR documentation system.

### 4.15. Statistical Analysis

All the data were presented as mean ± SD. Each experiment was carried out in triplicates. For the MWM test, escape latency times in the hidden platform trial were analyzed via two-way ANOVA of repeated measures. Statistical differences were analyzed by one-way ANOVA in combination with post hoc Tukey’s test (α = 0.05) to assess the difference between any two groups by using GraphPad Prism 8.0 statistical software (GraphPad Software, Inc., San Diego, CA, USA). *p* < 0.05 and *p* < 0.01 were considered statistically significant.

## Figures and Tables

**Figure 1 ijms-23-06354-f001:**
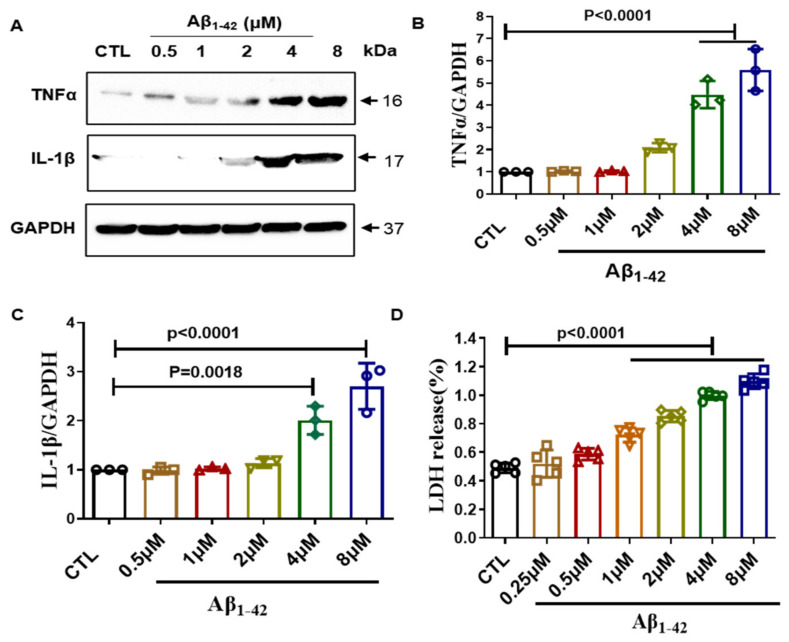
Establishment of cell inflammation model in BV2 cells. (**A**) BV2 cells were treated with different doses of Aβ_1–42_ (0.125–8 μM), and the expression of TNF-α and IL-1β were detected by using Western blot. (**B**,**C**) Quantification of TNF-α and IL-1β results in A. (**D**) BV2 cells were treated with different doses of Aβ_1–42_ (0.125–8 μM), and the cell membrane damage was measured by LDH assay. *p* < 0.05 was considered significantly different.

**Figure 2 ijms-23-06354-f002:**
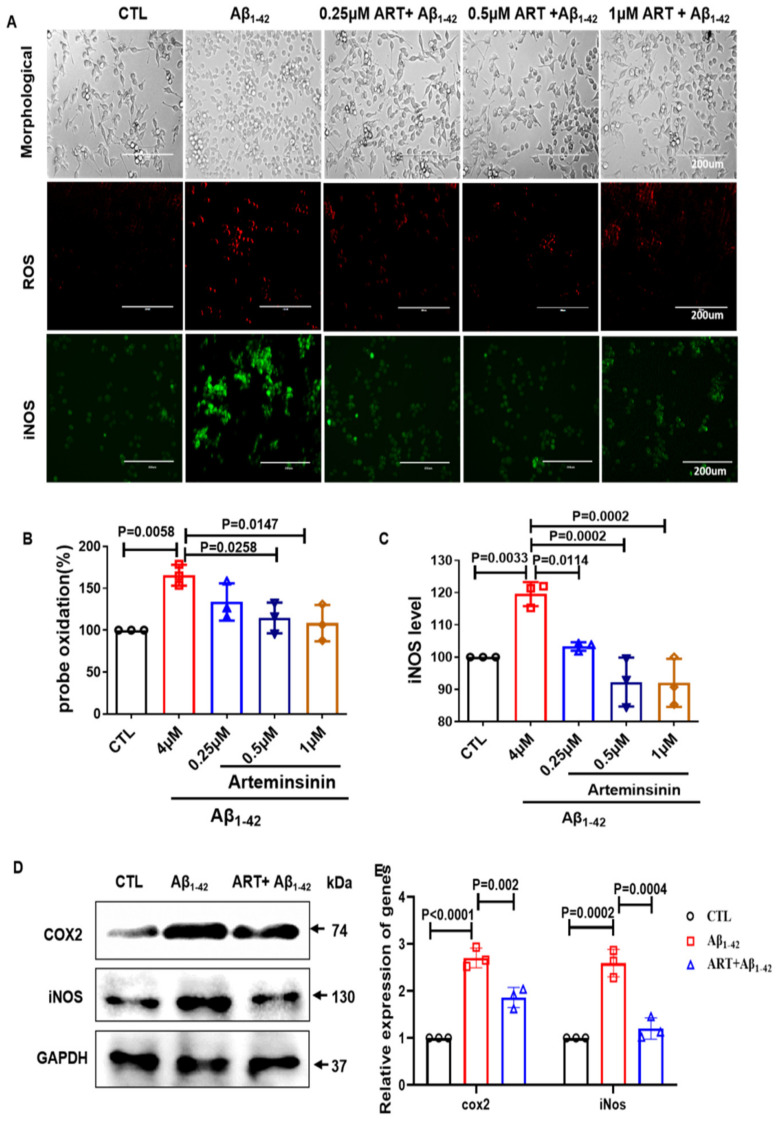
Artemisinin attenuated the oxidative stress induced by Aβ_1–42_ in BV2 cells. (**A**) BV2 cells were pretreated with different doses of Artemisinin (0.25–1 μM) before exposing them to 4 μM Aβ_1–42_, then morphological changes of microglia and the level of ROS and iNOS of BV2 cells were checked. (**B**) Quantification of ROS staining. (**C**) Quantification of iNOS levels. (**D**) Expression of COX2 and iNOS were detected by using Western blot. (**E**) Quantification of COX2 and iNOS. *p* < 0.05 was considered significantly different.

**Figure 3 ijms-23-06354-f003:**
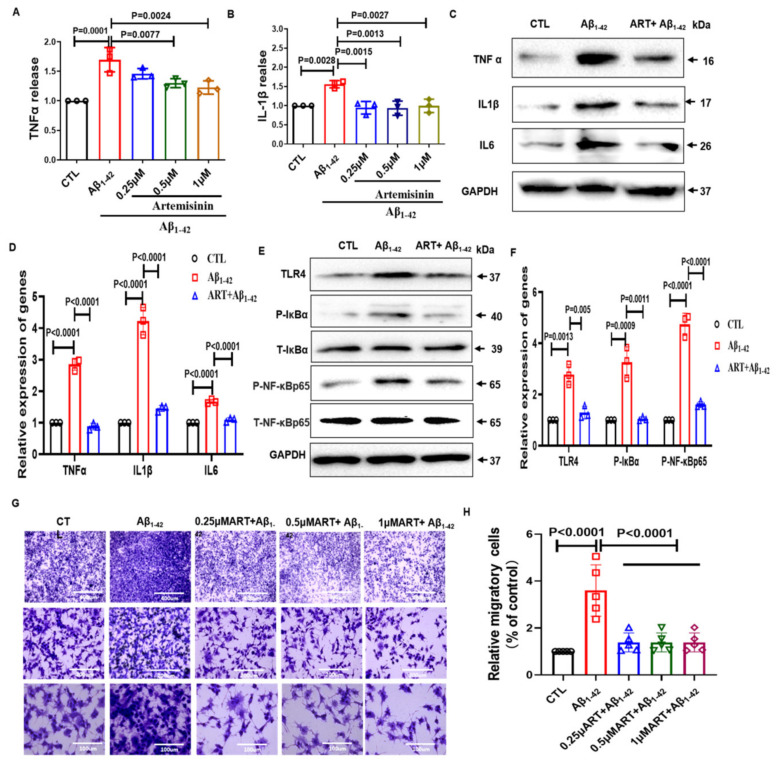
Artemisinin reduced the release of inflammatory factors and inhibited the migratory ability of BV2 cells caused by Aβ_1–42_. (**A**) ELISA assay checks the effect of Artemisinin on the TNF-α release induced by Aβ_1–42_. (**B**) ELISA assay checks the effect of Artemisinin on the IL-1β release induced by Aβ_1–42_. (**C**) The expression of inflammatory factors was detected by Western blot. (**D**) Quantification of inflammatory factors expression in C. (**E**) Western blot of p-NF-κB and p-IκBα in Aβ induced BV2 cells. (**F**) Quantitation of p-NF-κB, p-IκBα in (**E**). (**G**) Transwell chamber migration of BV2 microglial cells. (**H**) The relative migratory cells in the lower chamber. *p* < 0.05 was considered significantly different.

**Figure 4 ijms-23-06354-f004:**
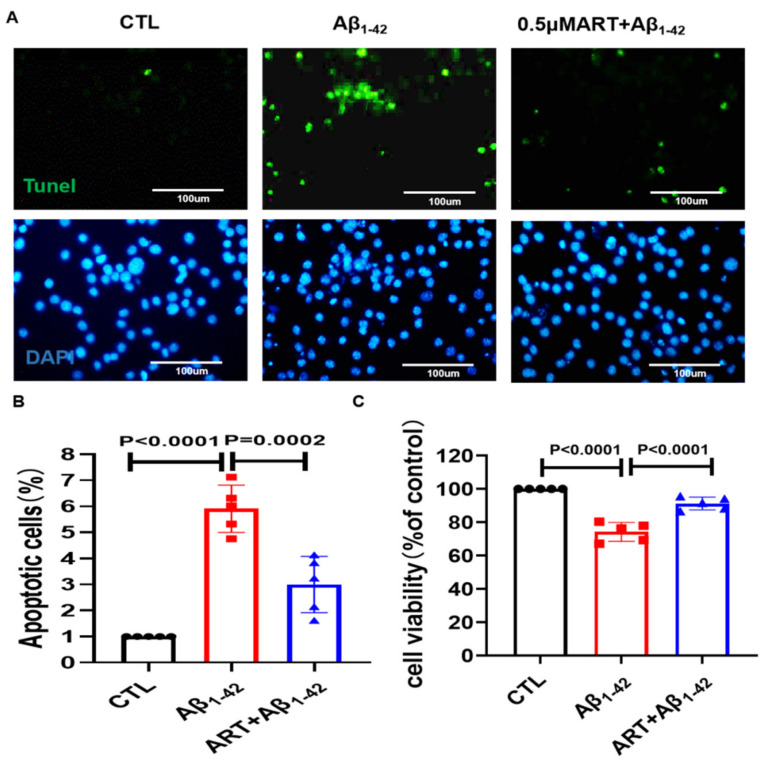
Artemisinin-pretreated microglia medium reduced PC12 cell damage. (**A**) Tunel staining was performed to check the protective effect of Artemisinin pretreated microglia medium on PC12 cells. (**B**) Quantification of apoptotic cells in (**A**). (**C**) Cell viability was detected by MTT assay. *p* < 0.05 was considered significantly different.

**Figure 5 ijms-23-06354-f005:**
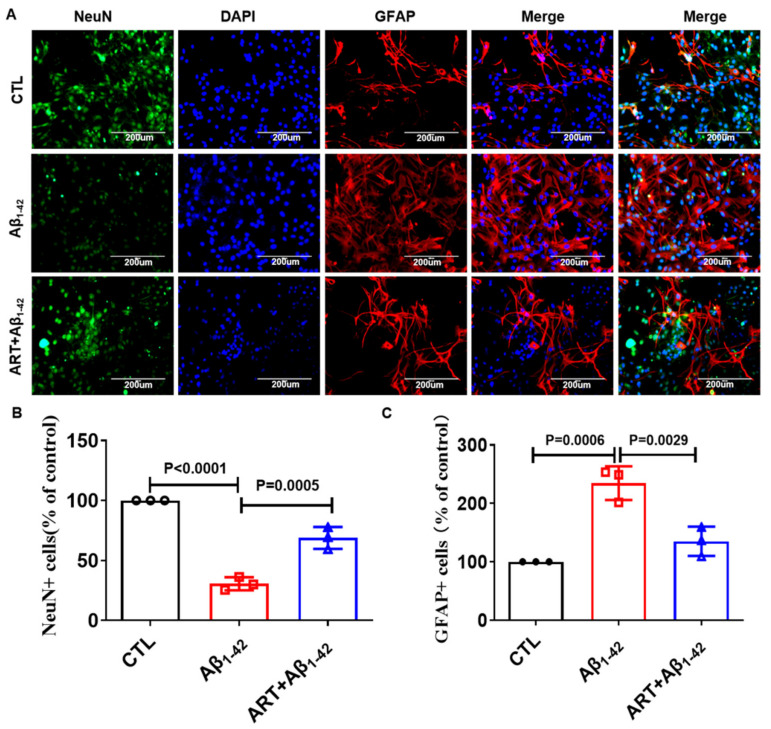
Artemisinin reduced the excessive activation of astrocytes in Primary cultured neuronal cells. (**A**) IF of NeuN and GFAP in primary cultured neuronal cells. (**B**) Quantitation of NeuN+ cells in A. (**C**) Quantitation of GFAP+ cells in A. *p* < 0.05 was considered significantly different.

**Figure 6 ijms-23-06354-f006:**
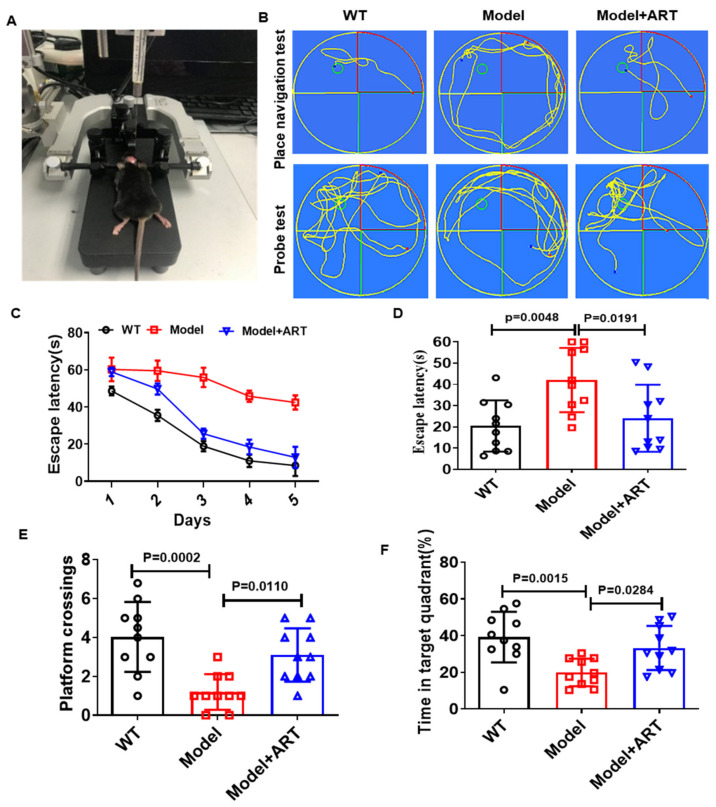
Artemisinin treatment improved the learning and memory of AD model mice. (**A**) Schematic of the experimental model. (**B**) The representative curve of probe test and place navigation test. (**C**,**D**) Time is required to find the hidden platform. (**E**) The number of times each group of mice crossed the hidden platform. (**F**) The percentage of time spent in the target quadrant of each group of mice. *p* < 0.05 was considered significantly different.

**Figure 7 ijms-23-06354-f007:**
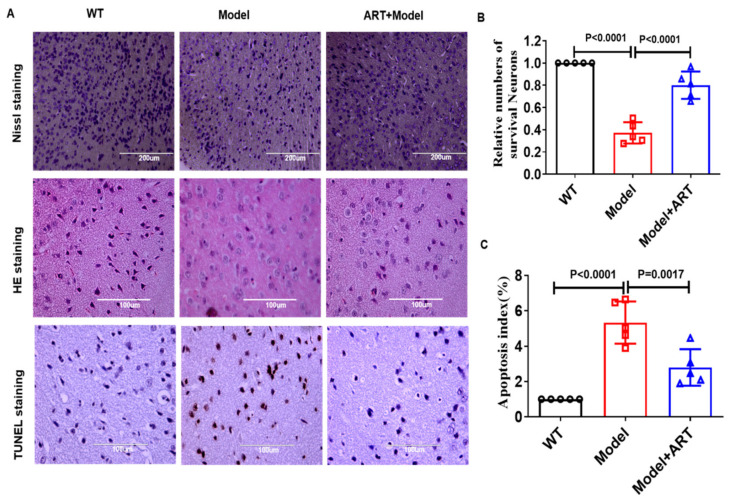
Artemisinin treatment reduced cell damage in the AD mouse model. (**A**) HE staining showed the histopathological changes of neurons in C57 model mice; Nissl staining was used to check the apoptosis of the Nissl body; TUNEL staining was used to check the apoptosis of neuronal cells in different groups. (**B**) Statistical analysis results of Nissl staining. (**C**) Statistical analysis results of TUNEL staining. *p* < 0.05 was considered significantly different.

**Figure 8 ijms-23-06354-f008:**
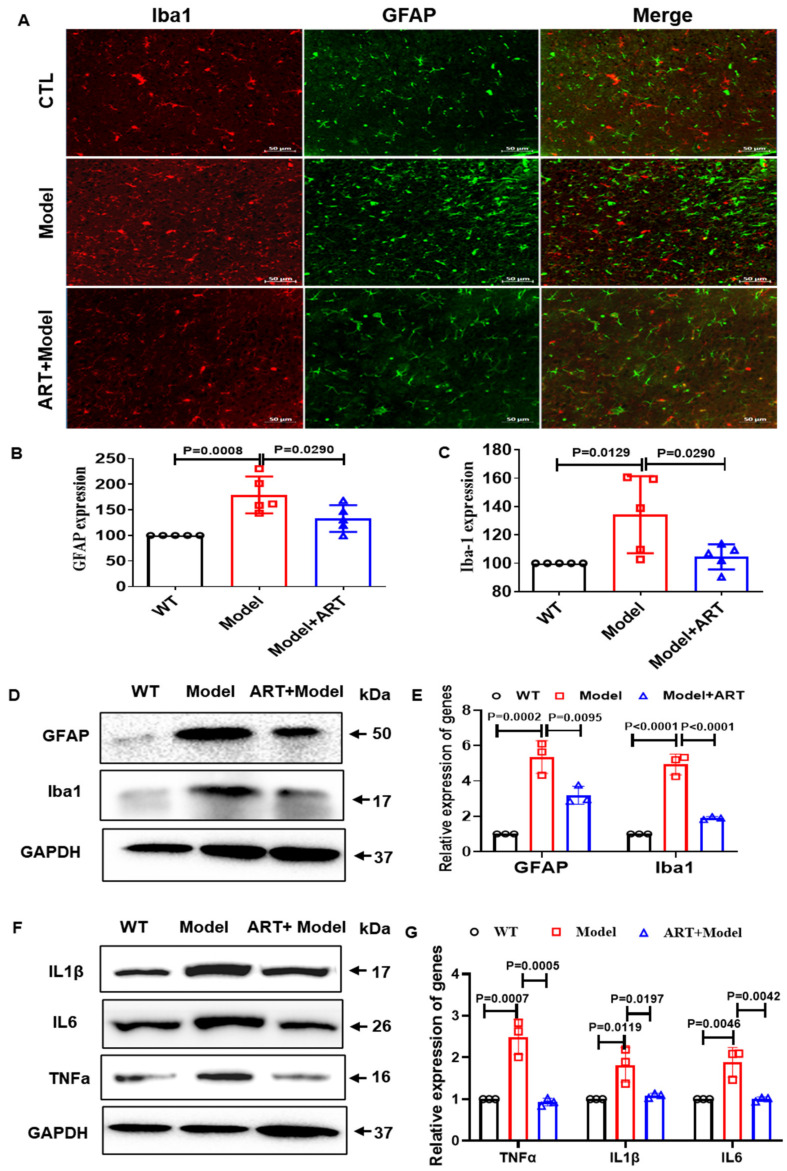
Artemisinin improves the excessive activation of glial cells. (**A**) IF of GFAP and Iba-1 in different groups of C57 model mice (**B**,**C**) Quantitation the expression level of GFAP and Iba1. (**D**) Western blot of GFAP and Iba-1 in C57 model mice. (**E**) Quantitation of the expression level of GFAP and Iba1 in (**D**). (**F**) Artemisinin reduced the release of inflammatory factors. (**G**) Quantitation of the expression level of TNF-α, IL-1β and IL-6 in (**F**). *p* < 0.05 was considered significantly different.

**Figure 9 ijms-23-06354-f009:**
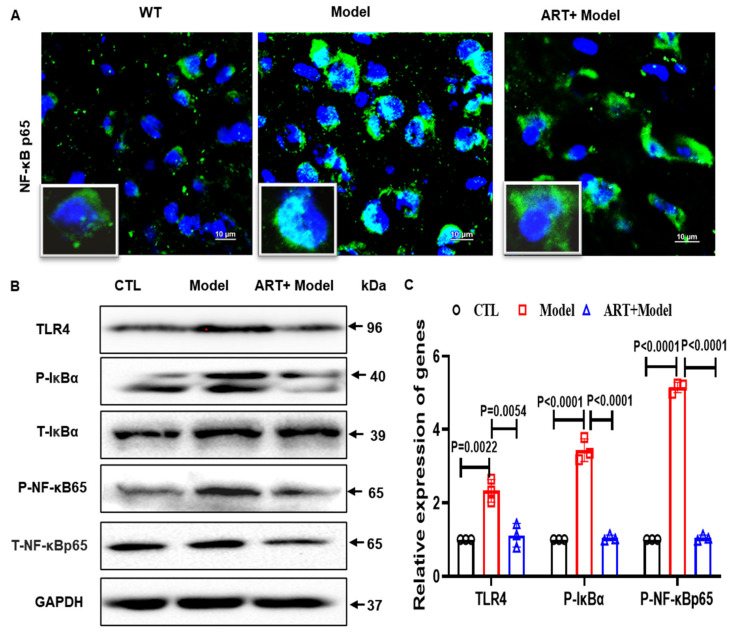
Artemisinin inhibits TLR4/NFκB signaling pathway. (**A**) IF of NF-κB65 in the C57 model. (**B**) Western blot of TLR4, p-NF-κB and p-IκBα in the C57 model. (**C**) Quantitation the expression level of TLR4, p-NF-κB and p-IκBα in (**A**). *p* < 0.05 was considered significantly different.

**Figure 10 ijms-23-06354-f010:**
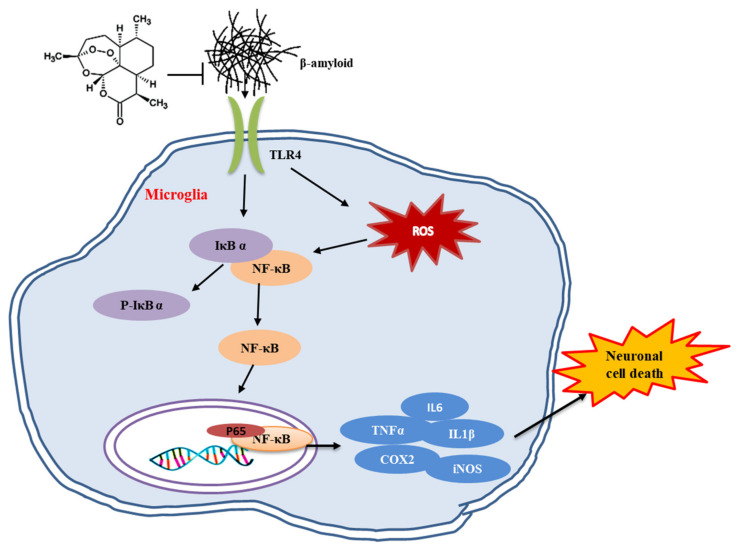
The possible mechanism of Artemisinin against neuroinflammation in AD. Aβ_1–42_ stimulated microglial receptor TLR4 to activate the IκBα/NF-κB signaling pathway, resulting in the massive expression of intracellular inflammatory factors (IL-6, TNF-α, IL-1β) and an increase in the level of ROS(COX2, iNOS). While Artemisinin pretreatment suppressed the activation of the TLR4/IκBα/NF-κB pathway and reduced the release of inflammatory factors, which in turn protects nerve cells from damage by improving the inflammatory environment in the brain.

**Table 1 ijms-23-06354-t001:** Antibody information.

Antibody	Cat. NO	Company	Dilution
TLR4	35463	SAB	WB:1:1000
NF-κB p65	4764	CST	WB:1:1000
P-NF-κB p65	3033	CST	WB:1:1000
IκBα	4814	CST	WB:1:1000
P- IκBα	2859	CST	WB:1:1000
GFAP (GA5)	3670	CST	WB:1:1000/IF:1:200
TNF-α	41504	SAB	WB:1:1000
IL-6	32064	SAB	WB:1:1000
IL-1β (3A6)	12242s	CST	WB:1:1000
Cox2	21679	SAB	WB:1:1000
iNOS	13120	CST	WB:1:1000
Iba1	49668	SAB	WB:1:500/IF:1:1000
Anti-rabbit IgG HRP	7074	CST	WB:1:2000
Alexa Fluor^®^ 594	8889	CST	IF:1:500
Alexa Fluor^®^ 488	4412	CST	IF:1:500

**Table 2 ijms-23-06354-t002:** Detail information of β-Amyloid (1–42).

Sequence (Three-Letter Code)	H-Asp-Ala-Glu-Phe-Arg-His-Asp-Ser-Gly-Tyr-Glu-Val-His-His-Gln-Lys-Leu-Val-Phe-Phe-Ala-Glu-Asp-Val-Gly-Ser-Asn-Lys-Gly-Ala-Ile-Ile-Gly-Leu-Met-Val-Gly-Gly-Val-Val-Ile-Ala-OH
One Letter Code	DAEFRHDSGYEVHHQKLVFFAEDVGSNKGAIIGLMVGGVVIA

Molecular Formula: C_203_H_311_N_55_O_60_S, Molecular Mass: 4514.10.

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
