# Peer review of "Artemisinin Attenuates Amyloid-Induced Brain Inflammation and Memory Impairments by Modulating TLR4/NF-κB Signaling"

_ijms, 2022, doi:10.3390/ijms23116354_

Round 1
Reviewer 1 Report
This manuscript by Xia Zhao et al. describes the effects of artemisinin in in vitro and in vivo models of Alzheimer's disease (AD). After careful review of this manuscript, I strongly advise them to revise and resubmit their study after making the changes suggested in the following comments.
Generally, the abstract should be rewritten. It is not informative and does not reflect the aim of the research and the most important findings of the study.
Please add p-value in the abstract when mentioned the word significant.
Abbreviations should be defined in the first time they are mentioned.
In the introduction, the authors should work hard in revising the manuscript to increase its quality to the level of publication. Some of the major flaws include, the lack of organization in the presented information, there are many grammatical errors, while there are also many statements are without corresponding citation lines 52, 53, 54-57,60-61....etc
At the end of the introductory section, the authors should explicitly state the study's rationale and purpose.
The authors should not talk about their results in the introduction section. They should remove this statement line 75-78.
The authors must indicate the exact P-value based on the results of statistical analysis achieved in the result section, as well as the number of experiment repetitions for in vitro and the number of animals for in vivo studies.
The important findings of the study should be discussed first in the discussion section. Furthermore, the data presented appears to be inadequate and unconvincing. Many statements are likewise devoid of supporting citations. I strongly encourage the writers to revise the discussion section in light of the findings.
In materials and methods section, the authors are required to provide the specific animal model for the study (species, age , number and gender ).
How many animals are used for Stereotaxic injection??
How many animals were used for behavioral studies??
Blood samples from the retro-orbital sinus in mice not a suitable method for blood collection
Author Response
Response to Reviewer 1 Comments
Point 1:Generally, the abstract should be rewritten. It is not informative and does not reflect the aim of the research and the most important findings of the study.Please add p-value in the abstract when mentioned the word significant.
Response 1: Thanks a lot for your comments. We have rewritten our abstract carefully as you suggested. Please see the revised manuscript.
Point 2:Abbreviations should be defined in the first time they are mentioned.
Response 2: Thanks a lot for your comments. We have done as suggested.
Point 3:In the introduction, the authors should work hard in revising the manuscript to increase its quality to the level of publication. Some of the major flaws include, the lack of organization in the presented information, there are many grammatical errors, while there are also many statements are without corresponding citation lines 52, 53, 54-57,60-61....etc. At the end of the introductory section, the authors should explicitly state the study's rationale and purpose. The authors should not talk about their results in the introduction section. They should remove this statement line 75-78.The authors must indicate the exact P-value based on the results of statistical analysis achieved in the result section, as well as the number of experiment repetitions for in vitro and the number of animals for in vivo studies.
Response 3: We thank and appreciate the reviewer's suggestion. According, we have revised our manuscript carefully.
- We have carefully modified the language and added the missing reference as the reviewer suggested.
- According to the reviewer'scomment, we have improved the introduction. Moreover, we removed our results in the introduction as suggested. Please see the revised manuscript.
- We have carefully modified the result section. We added the exact P-value as you suggested. The number of experiment repetitions is shown as a scatter in the figure. Please see the revised manuscript.
Point 4:The important findings of the study should be discussed first in the discussion section. Furthermore, the data presented appears to be inadequate and unconvincing. Many statements are likewise devoid of supporting citations. I strongly encourage the writers to revise the discussion section in light of the findings.
Response 4: Thanks a lot for your valuable comments. We have carefully modified the discussion section and added the supporting reference. Please see the revised manuscript.
Point 5:In materials and methods section, the authors are required to provide the specific animal model for the study (species, age , number and gender ). How many animals are used for Stereotaxic injection?? How many animals were used for behavioral studies?? Blood samples from the retro-orbital sinus in mice not a suitable method for blood collection
Response 5: We thank and appreciate the reviewer for touching this point. We have added animal species, age, number, and gender in our manuscript 4.3 Animal and treatment. In total, we used 30 mice for Stereotaxic injection and behavioral studies. Please see the revised manuscript. Thank you for pointing out the problem with our eye blood collection, we will improve our blood collection method in future experiments.

Reviewer 2 Report
The authors propose a complex study on the biological activity of artemisinin in-vitro and in-vivo. In particular, the authors use artemisinin to counteract the neuro-inflammation related to cognitive neuro degeneration. For this reason they measure the effects on the cognitive performances of mice injected with b-amylode peptide and subsequently treated with artemisinin. They also perform immunohistochemistry to assess the degree of expression of markers of cognitive impairment.
The results collected are promising and could incentivize further pharmaceutical chemistry studies to improve the artemisinin molecule as therapeutic strategy.
Furthermore, the study is original because there are few works that use artemisinin in in-vitro and or in-vivo models of neuro degeneration.
However, I found the introduction a bit difficult to read. There are several typos. Also the references should be updated. In this regard, see:
Author Response
Response to Reviewer 2 Comments
Point:The authors propose a complex study on the biological activity of artemisinin in-vitro and in-vivo. In particular, the authors use artemisinin to counteract the neuro-inflammation related to cognitive neuro degeneration. For this reason they measure the effects on the cognitive performances of mice injected with b-amylode peptide and subsequently treated with artemisinin. They also perform immunohistochemistry to assess the degree of expression of markers of cognitive impairment.
The results collected are promising and could incentivize further pharmaceutical chemistry studies to improve the artemisinin molecule as therapeutic strategy.
Furthermore, the study is original because there are few works that use artemisinin in in-vitro and or in-vivo models of neuro degeneration.
However, I found the introduction a bit difficult to read. There are several typos. Also the references should be updated.
Response: We thank and appreciate the reviewer's comment. We carefully modified our manuscript as suggested. Also, we have updated the references. Please have a look at the revised manuscript.

Round 2
Reviewer 1 Report
Although the authors responded to the majority of the comments, some remained unanswered, thus I urge the writers to edit the paper and make the following changes.
In the introduction section, still, the authors talk about their results .... line 83-84
Several places in the results section did not include the exact P-value, for example line 92 and many other places.
The authors should add the molecular weight of the proteins in figures with western blotting results.
The number of animals used for stereotaxic injection has yet to be determined, as well as how many animals perished as a result of this treatment.
Many statements in the discussion are likewise devoid of supporting citations, for example, lines 256, 257, 264, 265, and other places. Still, the discussion requires some improvements
Author Response
Response 1: We thank and appreciate for pointing this out. We have included your suggestion in the revised manuscript. In order to facilitate your examination, all changes we made in the manuscript have been highlighted in blue.
- As you suggested, we have removed the description of the results (line 83-84) in the introduction. Please see the revised manuscript.
- We have already added exact P-values in the figures and we also explained when P < 0.05 was considered significantly different in each figure legends. In the results section, we sometimes analyze a class of indicators at the same time, such as the expression of related inflammatory factors. Such precise P-values are difficult to properly add, So, we all added in Figure and Figure legends.
- We have added the molecular weight of the proteins in figures with western blotting results as you suggested. Please see the revised manuscript.
- As we described, a total of 30 mice were used for the experiment. The 30 mice were divided into three groups: WT, Model, Artemisinin+Model;A total of 20 mice in the model group and Artemisinin + model group were injected with Aβ1-42 (10μg) respectively, and the WT group was injected with the same volume of PBS. No mice died during the experiment. Please see the Materials and Methods 4.2 and3 in our revised manuscript.
- We carefully checked the statements in the discussion you mentioned. Some of them (256,257) are descriptions of the results obtained in our article and are not cited from other articles, so we did not insert references. We have added the reference of lines 264, 265 and other places as you suggested. In addition, we have improved our discussion section. Please see the revised manuscript.
